Anti-ENO1 antibody combined with metformin against tumor resistance: a novel antibody-based platform

Shu Xiong 1
Zhang Hui Wen 2
Liu Shi Ya 2
Sun Li Xin 2
Zhang Tao 3
Ran Yu Liang 2 ranyuliang@cicams.ac.cn
1 National Center for Orthopaedics, Beijing Research Institute of Traumatology and Orthopaedics, Beijing JiShuiTan Hospital , Beijing , China
2 State Key Laboratory of Molecular Oncology, National Cancer Center/National Clinical Research Center for Cancer/Cancer Hospital, Chinese Academy of Medical Sciences and Peking Union Medical College , Beijing , China
3 The Second People’s Hospital of Xining , Xining , China
Uversky Vladimir
Electronic publication date: 2024 Mar 18
Publication date: 2024
Volume: 12
Electronic Location ID: e16817
Received 2023 Mar 31; Accepted 2023 Dec 30
Copyright: © 2024 Shu et al.
Copyright year: 2024
Copyright holder: Shu et al.
License: This is an open access article distributed under the terms of the Creative Commons Attribution License, which permits unrestricted use, distribution, reproduction and adaptation in any medium and for any purpose provided that it is properly attributed. For attribution, the original author(s), title, publication source (PeerJ) and either DOI or URL of the article must be cited.
License URL: https://creativecommons.org/licenses/by/4.0/

Keywords: Antibody-based drug combination therapy, Cancer stem cells, Drug resistance, ENO1, Oncotherapy

Funding: CAMS Innovation Fund for Medical Sciences 2021-I2M-1-067 National Natural Science Foundation of China 82073278 Beijing Natural Science Foundation 7222012 Beijing Municipal Health Commission BMHC-2019-9 and BMHC-2021-6 State Key Laboratory of Molecular Oncology SKLMO-2021-17 Qinghai Provincial Science and Technology Department Applied Basic Research Program 2020-ZJ-779 The study was supported by the CAMS Innovation Fund for Medical Sciences [grant number, 2021-I2M-1-067]; the National Natural Science Foundation of China [grant number, 82073278]; the Beijing Natural Science Foundation [grant number, 7222012]; Beijing Municipal Health Commission [grant number, BMHC-2019-9, and BMHC-2021-6]; the Independent Issue of State Key Laboratory of Molecular Oncology [grant number, SKLMO-2021-17]; and the Qinghai Provincial Science and Technology Department Applied Basic Research Program [grant number, 2020-ZJ-779]. The funders had no role in study design, data collection and analysis, decision to publish, or preparation of the manuscript.

==============================
Background

Antibody-based platforms (i.e., ADC) have emerged as one of the most encouraging tools for the cancer resistance caused by cancer stem cells (CSCs) enrichment. Our study might provide a promising therapeutic direction against drug resistance and serve as a potential precursor platform for screening ADC.

Methods

The cell migration, invasion, drug resistance, and self-renewal were assessed by the cell invasion and migration assay, wound healing assay, CCK-8 assay, colony formation assay, and sphere formation assay, respectively. The expression profiles of CSCs (ALDH+ and CD44+) subpopulations were screened by flow cytometry. The western blot and cell immunofluorescence assay were used to evaluate pathway-related protein expression in both anti-ENO1 antibody, MET combined with DPP/CTX-treated CSCs.

Results

In the present study, western blot and flow cytometry verified that anti-ENO1 antibody target the CD44+ subpopulation by inhibiting the PI3K/AKT pathway, while metformin might target the ALDH+ subpopulation through activation of the AMPK pathway and thus reverse drug resistance to varying degrees. Subsequently, in vitro investigation indicated that anti-ENO1 antibody, metformin combined with cisplatin/cetuximab could simultaneously target ALDH+ and CD44+ subpopulations. The combination also inhibited the CSCs proliferation, migration, invasion, and sphere formation; which may result in overcoming the drug resistance. Then, molecular mechanism exploration verified that the anti-ENO1 antibody, metformin combined with cisplatin/cetuximab inhibited the Wnt/β-catenin signaling.

Conclusions

The study preliminarily revealed anti-ENO1 antibody combined with metformin could overcome drug resistance against CSCs by inhibiting the Wnt//β-catenin pathway and might serve as a potential precursor platform for screening ADC. More importantly, it is reasonably believed that antibody-based drug combination therapy might function as an encouraging tool for oncotherapy.

Significance Statement

Our study preliminarily revealed that anti-ENO1 antibody plus metformin could overcome drug resistance against CSCs by inhibiting the Wnt//β-catenin pathway and might serve as a potential precursor platform for screening ADC. More importantly, it is reasonably believed that antibody-based drug combination therapy might function as an encouraging tool for oncotherapy.

Introduction

Cancer is one of the leading causes of death worldwide (Phi et al., 2018). Although great breakthroughs have been made in cancer over the past few decades, traditional treatment methods such as chemotherapy are still the mainstay of cancer treatment (Wang, Zhang & Chen, 2019). However, multiple drug resistance (MDR) may result in treatment failure or cancer recurrence, even with ≥90% deaths in patients receiving conventional chemotherapy or new targeted drugs (Bukowski, Kciuk & Kontek, 2020). Rapidly increasing studies are focused on avoiding or reversing drug resistance. Among them, cancer stem cells (CSCs) are considered to be the cause of drug resistance, chemo- and radiotherapy resistance, and cancer relapse, as they have the characteristics of inducing cell cycle arrest, heterogeneity, and self-renewal (Eun, Ham & Kim, 2017; Najafi, Mortezaee & Majidpoor, 2019; Cojoc et al., 2015). Accumulating evidence indicates that following chemotherapy of any tumor, there are residual CSCs (higher invasiveness and resistance) enriched (Brown, Hua & Tanwar, 2019; Quayle, Ottewell & Holen, 2018). Therefore, increasing anti-tumor therapeutics focus on targeting CSCs to provide better therapeutic effects and rationales (Barbato & Bocchetti, 2019; Phi et al., 2018; Yang et al., 2020). Among them, the antibody could serve as a platform biomaterial to participate in cancer treatment against CSCs together with small-molecule drugs, such as antibody-drug conjugates (ADC), which have emerged in the few decades as one of the most encouraging tools for the selective ablation of cancer cells (Marcucci et al., 2019). However, for ADC, the production of humanized monoclonal antibody (mAb) and subsequent drug conjugation are costly, and time-consuming (Zhao et al., 2015). Hence, developing antibody-based drug combination therapy against CSCs to reverse traditional therapeutic resistance and possibly function as a precursor platform for ADC screening remains crucial, but relevant studies are few and worthy of further exploration.

Given the above theory of the heterogeneity of CSCs, other studies have further found that there are two widely convertible CSCs subpopulations (epithelial CSCs with strong proliferation and self-renewal (ALDH+ subpopulation), and mesenchymal CSCs with strong migration and invasion (CD44+ subpopulation)) in response to different metabolic pathways and therapeutic agents (Sancho, Barneda & Heeschen, 2016; Shibue & Weinberg, 2017). The transition, also known as plasticity or epithelial-to-mesenchymal transition (EMT), prompts cancer cells to undergo phenotypic microevolution adapted to the microenvironment, ultimately driving cancer progression and recurrence (Shlyakhtina & Moran, 2021). Increasing evidence demonstrated that conventional therapies often fail to eradicate cancer cells entering the CSC state by activating the EMT program, thus allowing CSC-mediated clinical relapse (Shibue & Weinberg, 2017). Therefore, we proposed that CSCs might be eliminated to further reverse drug resistance by simultaneously targeting two subpopulations. Previous studies have indicated that therapeutic monoclonal antibodies against CSCs when combined with chemotherapy may highlight the potential of improving anti-tumor effects and be an attractive option for a new therapeutic approach (Sneha et al., 2017; Yang et al., 2020; Du et al., 2013; Yang et al., 2021).

Enolase 1 (ENO1) is a subtype of enolase, which anchored on the cell membrane and served to greatly activate plasminogen to stimulate the migration and invasion ability of tumor cells (Qiao et al., 2021). Thus, ENO1 was considered an important target in cancer treatment and acted as a good prognostic indicator to monitor the disease progression (Cheng et al., 2019; Principe et al., 2017). Of note, recent studies have revealed that anti-ENO1 antibody was associated with the CD44+ subpopulation, blocked the malignant transformation of CSCs (Huang et al., 2021; Principe et al., 2015; Shu et al., 2021), and prevented cancer metastasis and prolonged survival (Li et al., 2021). Besides, metformin (MET) is considered an effective small-molecule drug for regulating ALDH+ subpopulation and preventing CSCs-driven chemoresistance (Brown et al., 2020). Taken together, it was hypothesized that anti-ENO1 antibody combined with MET might possess synergistic interaction to eliminate CSCs completely and further reverse drug resistance (Lv & Shim, 2015; Zhu et al., 2015).

ALDH+ and CD44+ CSCs subpopulations are often expressed in gastric cancer (GC) and non-small cell lung cancer (NSCLC) (Masciale et al., 2020; Nishikawa et al., 2013). Moreover, GC and epidermal growth factor receptor (EGFR)-mutant NSCLC are still serious public health problems in China (the incidence is far higher than in Europe and America) (Sung, Ferlay & Siegel, 2021; Xu et al., 2020). Thus, we selected these two cell lines for subsequent studies. As is well-known, cisplatin (DDP)-based systemic chemotherapy as first-line treatment for GC always developed resistance after a few cycles of treatment, further limiting the overall clinical efficacy (Huang et al., 2019). On another side, cetuximab (CTX) is currently the focus of intense investigation in various NSCLC populations (Reade & Ganti, 2009). Although CTX is currently used more as a combination therapy, it also offers therapeutic options for patients who are resistant to the first- or second-EGFR-tyrosine kinase inhibitors (EGFR-TKIs) (Mazzarella, Guida & Curigliano, 2018; Reade & Ganti, 2009). Of note, drug resistance to CTX in other cancers also raises concerns (Zhou, Ji & Li, 2021). The drawbacks of two conventional remedies further support the exploration of reversing drug resistance by selecting the above two cancer cell lines.

Taken together, we focused on these two cancer cell lines and proposed an antibody-based drug combination therapy, that is, the anti-ENO1 antibody combined with MET could reverse drug resistance caused by CSCs in conventional treatments (DDP or CTX), and preliminarily explored the mechanism. In parallel, we believe that antibodies combined with small-molecule drugs might provide a promising direction for drug repositioning and other cancer treatment against drug resistance, and anti-ENO1 antibody combined with MET might serve as a potential precursor platform for screening ADC.

Materials and Methods

Cell lines and culture

Human GC cell lines PAMC82 and human NSCLC cell lines A549 were obtained from the Laboratory of Antibody Engineering, Cancer Institute, Chinese Academy of Medical Sciences (Beijing, China). PAMC82 and A549 cells were cultured in Roswell Park Memorial Institute (RPMI)-1,640 medium (Gibco, Grand Island, NY, USA) containing 10% fetal bovine serum (Kang Yuan Biology, Dailan, China) and 1% penicillin/streptomycin solution (North China Pharmaceutical Company, Shijiazhuang, China) in an air atmosphere containing 5% CO2 at 37 °C. Then, all cells were subcultured using 0.1% EDTA and 0.2% trypsin when confluent (>80% confluence).

Cell counting kit-8 (CCK-8) assay for drug susceptibility

Briefly, 5 × 103 PAMC82 and A549 cells cultured in above-mentioned medium per well were respectively seeded into a 96-well plate. The cells were treated with different drugs for 72 h after cell adherence. Four drugs (DDP (Shandong, China), CTX (Merck Serono, Aubonne, Switzerland), MET (Sigma, Burlington, MA, USA), and anti-ENO1 antibody (18H12; the Laboratory of Antibody Engineering, Cancer Institute, Chinese Academy of Medical Sciences, China) were used. Then, 100 μL of CCK-8 reagent (Dojindo Laboratories) was added to each well for 2 h at 37 °C. The absorbance at 450 nm was measured using a microplate reader (SpectraMax M5; Sunnyvale, CA, USA) for drug susceptibility. The experiment was carried out in three repeated holes, and three parallel experiments were performed for each sample.

Cell counting kit-8 (CCK-8) assay for drug susceptibility

Cells treated with monotherapy or combination of DDP, CTX, MET, and anti-ENO1 antibody for 96 h were seeded in 96-well plates (1,500 cells/well) and cultured for 2 h in the serum-containing media. Then, the absorbance was measured at 450 nm at 24, 48, 72, and 96 h by a microplate reader (SpectraMax M5; Sunnyvale, CA, USA) for cell proliferation potential. The experiment was carried out in three repeated holes, and three parallel experiments were performed for each sample.

Cell proliferation ability assay

PAMC82 and A549 cells were sub-cultured at >80% confluence in the 10-cm dish under a 5% CO2 air atmosphere at 37 °C. After adherence, the cells were co-cultured with monotherapy or combination of DDP, CTX, MET, and anti-ENO1 antibody in the serum-containing media for 96 h. Subsequently, the number of viable cells was evaluated using a cell counter. Five parallel experiments were performed for each sample.

Sphere formation assay

Self-renewal ability was evaluated by sphere formation assay. The cells treated with monotherapy or combination of DDP, CTX, MET, and anti-ENO1 antibody for 96 h were seeded in 24-well ultra-low attachment plates (500 cells/well) and cultured in Dulbecco’s Modified Eagle Medium/Nutrient Mixture F12 (DMEM/F12) supplemented with 0.8% methylcellulose (Sigma, Burlington, MA, USA) and growth factor (epidermal growth factor, basic fibroblast growth factor, B27, and leukemia inhibitory factor). The cells were cultured at 37 °C in 5% CO2. Each group was carried out in three repetitive holes and the spheroidization was observed on days 3, 5, 7, 9, and 11 respectively. Three parallel experiments were performed for each sample. Spheres generated were photographed and sphere numbers were counted under a light microscope.

Wound healing assay

The cells treated with monotherapy or combination of DDP, CTX, MET, and anti-ENO1 antibody for 96 h were seeded at a density of 5 × 105 cells/well in 6-well plates and cultured at 37 °C till cell adherence. A wound was created using a sterile 2 μL pipette tip in each well. Images were taken at 0, 24, and 48 h using the EVOS FL microscope system and EVOS software (Life Technologies Inc, Carlsbad, CA, USA). ImageJ was used to quantify wound area. The experiment was carried out in two repeated holes, and three parallel experiments were performed for each sample.

Cell invasion and migration assay

For the cell invasion assay, the cells treated with monotherapy or combination of DDP, CTX, MET, and anti-ENO1 antibody for 96 h were seeded into the upper BioCoat Matrigel invasion chambers with 8.0 µm polyethylene terephthalate (PET) membrane (Corning, Corning, NY, USA). A complete medium containing 10% FBS was used as a chemoattractant in the lower compartment. The transwell chamber was maintained at 37 °C for 48 h; then, non-invading cells were washed using phosphate buffer solution (PBS) from the Matrigel upper surface. Invading cells were fixed by adding 4% paraformaldehyde for 30 min and stained with 0.1% crystal violet for another 30 min. The stained cells were counted under a light microscope (Nikon, Tokyo, Japan). Additionally, the cell migration assay was set up the same as the cell invasion assay except without Matrigel in chambers. Three parallel experiments were performed for each sample.

Flow cytometry

Protocol was followed according to the manufacturer (Stemcell Technologies, Vancouver, BC, Canada). Briefly, cells treated with monotherapy or combination of DDP, CTX, MET, and anti-ENO1 antibody for 96 h were resuspended in ALDEFLUOR assay buffer at a concentration of 5 × 105 cells/mL. Two tubes were labeled as control and sample. To the control tube, 5 µL of the DEAB inhibitor (ALDFLUORTMDEAB) was added. To the sample tube, 5 µL of the activated ALDEFLUOR reagent and 1 mL cell suspension were added and mixed; then, 500 µL of the suspension was taken out and put in the control tube with the inhibitor. Cells were incubated for 45 min at 37 °C, followed by the addition of 100 μL CD44 antibody (Abcam, Cambridge, UK) for another hour. Subsequently, cells were washed and resuspended with ALDEFLUOR buffer. ALDH+ and CD44+ subpopulations were profiled using a flow cytometer (BD FACS Aria, San Jose, CA, USA). Three parallel experiments were performed for each sample.

Western blot

The total protein was extracted using radio-immunoprecipitation assay (RIPA) lysis buffer (Applygen, Beijing, China) supplemented with the protease inhibitors cocktail (Roche, Basel, Switzerland). Protein concentrations were determined using BCATM Protein Assay kits (Thermo Fisher, Waltham, MA, USA). Equal amounts of cell lysates (15 μL per lane) were separated by sodium dodecyl sulfate-polyacrylamide gel electrophoresis (SDS-PAGE) and electro-transferred to PVDF membranes. After that, cell lysates were blocked with a blocking buffer for an hour. Subsequently, the membrane was exposed to various primary antibodies and incubated overnight at 4 °C. Afterward, the membrane was treated with the corresponding secondary antibodies conjugated with HRP. The bands were visualized using enhanced chemiluminescence and quantified. The obtained data were then normalized by comparing them to the density of β-actin, and ImageJ software (National Institutes of Health, Bethesda, MD, USA) was employed for quantification. Five parallel experiments were performed for each sample.

Colony formation assay

Collected cells treated with monotherapy or combination of DDP, CTX, MET, and anti-ENO1 antibody for 96 h were seeded (500 cells/dish) in 10-cm dishes. Then, the cells were cultured in 5% CO2 at 37 °C for 2 weeks till the macroscopic cloning had formed. Subsequently, the cells were fixed with 4% paraformaldehyde for 30 min and stained with 0.1% crystal violet (Solarbio Science & Technology Co., ltd, Beijing, China) for another half hour. Colony numbers were counted and images were taken under a light microscope (Nikon, Tokyo, Japan). Three parallel experiments were performed for each sample.

Cell immunofluorescence assay

The cells treated with monotherapy or combination of DDP, CTX, MET, and anti-ENO1 antibody for 96 h were inoculated into 24-well plates with a density of 1 × 105 cells/well and cultured until the fusion rate was 60–70%. For fixed-cell immunofluorescence, after washing with a serum-free medium, the cells went through fixation in 4% paraformaldehyde for 15 min followed by washing three times with PBS containing 0.1% bovine serum albumin (BSA) and 0.05% Tween-20. After that, the cells were permeable with 0.02% TritonX-100 for 10 min and 10% serum was utilized for the blocking (10 min). Then, the cells were later cultivated overnight with an anti-β-catenin primary antibody (CST, Danvers, MA, USA) at room temperature.

Statistical analysis

Data are presented as the mean ± standard error of the mean (SEM) for at least three independent experiments. Statistical analysis was conducted using GraphPad Prism software (version 8.3.0; GraphPad Software, Inc., La Jolla, CA, USA). To assess differences between two groups, t-tests were employed. For comparisons involving three or more groups, one-way ANOVA with Tukey’s post hoc test was utilized. Statistical significance was defined as p < 0.05.

Results

DDP and CTX produced drug resistance in GC and NSCLC cell lines

We examined the effects of DDP and CTX on cell growth in PAMC82 and A549 cell lines, respectively. The CCK-8 assay showed that the proliferation of both cells gradually decreased with increasing concentrations of DDP or CTX, and IC50 of PAMC82 and A459 cells were 0.4667 μg/ml and 35 μg/ml, respectively (Fig. 1A). We then tested the effects of two drugs on the CSC-like characteristics of both cells. The sphere formation assay showed that DDP or CTX did not significantly inhibit the self-renewal abilities of cancer cells (Figs. 1B and S1). Moreover, the wound healing assay demonstrated that the migration abilities of both cells were not markedly inhibited after DDP or CTX (Fig. 1C). Similarly, the cell invasion and migration assay showed that DDP or CTX had no significant inhibitory effects on the migration and invasion of cancer cells (Fig. 1D). Taken together, these data suggested that DDP or CTX had killing effects on cancer cells but have no significant inhibition on the CSC-like characteristics, that is, drugs were effective but produced drug resistance.

Figure 1 DDP and CTX inhibited the proliferation of GCs and NSCLCs, respectively, but did not significantly inhibit the stem-like characteristics of cancer cells, that is, drug resistance was developed.

(A) The CCK-8 assay was performed to detect the cell viability and half-maximal inhibitory concentrations of DDP-treated PAMC82 cells and CTX-treated A549 cells. (B) The sphere formation assay demonstrated that the self-renewal abilities of DDP-treated PAMC82 cells and CTX-treated A549 cells were both inhibited. Scale bar, 200 μm. (C) The wound healing assay demonstrated that the cell migration abilities of DDP-treated PAMC82 cells and CTX-treated A549 cells were both inhibited. Scale bar, 200 μm. (D) The cell invasion and migration assay demonstrated that the cell migration and invasion abilities of DDP-treated PAMC82 cells and CTX-treated A549 cells were both inhibited. Scale bar, 200 μm. Data are expressed as mean ± SEM.

ALDH+ and CD44+ CSCs subpopulations were upregulated by DDP; while ALDH+ subpopulation by CTX

The western blot for ABCG2 and TOP2A (stemness marker) characterized that PAMC 82 and A549 cells used were CSCs (Fig. S2). To further investigate the roles of the DDP and CTX in drug resistance, we evaluated the profiles of CSCs subpopulations by flow cytometry. As shown in Fig. 2A, DDP in PAMC82 cells could induce the significant up-regulation of the proportion of ALDH+ and CD44+ subpopulations. The proportion of the ALDH+ subpopulation was also remarkedly up-regulated after CTX in A549 cells while the CD44+ subpopulation was not significantly changed (Fig. 2B). Besides, we also found that DDP or CTX resulted in the increased expression of ABCG2 (drug efflux transporter) in all two subpopulations of CSCs compared with control group (Fig. S3). Based on these results, we reasonably suspected the increasing proportion of CSCs resulted in drug resistance such as metastasis and recurrence after treatment.

Figure 2 The aberrantly up-regulated ALDH+ and CD44+ subpopulations occurred with the DDP or CTX alone.

(A) Flow cytometry demonstrated that the proportions of ALDH+ and CD44+ subpopulations were up-regulated in DDP-treated PAMC82 cells. (B) Flow cytometry demonstrated that the proportion of the ALDH+ subpopulation was up-regulated in CTX-treated A549 cells. Data are expressed as mean ± SEM.

MET targeted ALDH+ subpopulation by activation of the AMPK pathway; while the anti-ENO1 antibody targeted the CD44+ subpopulation by inhibiting the PI3K/AKT pathway

We first explored the effects of cancer cell proliferation with MET and anti-ENO1 antibody. As shown in Fig. 3A, the proliferation of both cells gradually decreased with increasing concentrations of MET, and IC50 of PAMC82 and A459 CSCs were 12.85 and 7.283 mM, respectively. Similar results were also observed in the anti-ENO1 antibody group, MET combined with DDP or CTX group, and anti-ENO1 antibody combined with DDP or CTX group in both CSCs (Fig. 3A). Moreover, we verified that MET at 10 mM had the strongest inhibition on the self-renewal ability of PAMC82 CSCs, so the concentration of 10 mM was selected for subsequent assays (Fig. S4). The 18H12 was selected as the anti-ENO1 antibody for subsequent related studies due to the strongest binding affinity to ENO1 (Fig. S5). Moreover, as previously reported, the internalization of ENO1 antibody is not significant (Shu et al., 2021). We then explored the effects of MET and anti-ENO1 antibody on CSCs subpopulations and related characteristics. Flow cytometry showed that compared with DDP/CTX alone, MET combined with DDP/CTX significantly decreased the proportion of the ALDH+ subpopulation in two CSCs, while the CD44+ subpopulation was no statistically significant (Fig. S6). Compared with DDP/CTX alone, anti-ENO1 antibody plus DDP/CTX showed the reduced proportion of the ALDH+ and CD44+ subpopulations (Fig. S6). Sphere formation assay showed that MET combined with DDP/CTX decreased the self-renewal ability but anti-ENO1 antibody combined with DDP/CTX did not (Fig. 3B). The cell invasion and migration assay showed that MET combined with DDP could not inhibit the migration and invasion of PAMC82 CSCs, while anti-ENO1 antibody combined with DDP significantly did (Fig. 3C). Similar results were also observed in A549 CSCs, that is, MET combined with CTX could not inhibit the migration and invasion of A549 CSCs, while anti-ENO1 antibody combined with CTX significantly did (Fig. 3C). The western blot assay indicated that MET combined with DDP/CTX increased the expression of p-AMPK in two CSCs, while the anti-ENO1 antibody combined with DDP/CTX decreased the expression of p-AKT (Fig. 3D). Taken together, these data suggested that MET might target the ALDH+ subpopulation through activation of the AMPK pathway, while anti-ENO1 antibody target the CD44+ subpopulation by inhibiting the PI3K/AKT pathway, and thus reverse drug resistance to varying degrees.

Figure 3 MET targeted the ALDH+ subpopulation through activation of the AMPK pathway; while anti-ENO1 antibody targeted the CD44+ subpopulation by inhibiting the PI3K/AKT pathway.

(A) The CCK-8 assay was performed to detect the cell viability and proliferation of PAMC82 and A549 CSCs with MET and anti-ENO1 antibody. (B) The sphere formation assay demonstrated that the self-renewal abilities of PAMC82 and A549 CSCs were decreased with MET combined with DDP/CTX while anti-ENO1 antibody combined with DDP/CTX did not. Scale bar, 200 μm. (C) The cell invasion and migration assay demonstrated that MET combined with DDP/CTX did not inhibit the cell migration and invasion abilities of PAMC82 and A549 CSCs, while anti-ENO1 antibody combined with DDP/CTX did. Scale bar, 200 μm. (D) The western blot verified the expression of proteins related to the AMPK pathway in MET combined with DDP/CTX-treated PAMC82/A549 CSCs, and the expression of proteins related to the PI3K/AKT pathway in anti-ENO1 antibody combined with DDP/CTX-treated PAMC82/A549 CSCs. Data are expressed as mean ± SEM.

Anti-ENO1 antibody, MET combined with DDP/CTX could reverse drug resistance by simultaneously eliminating ALDH+ and CD44+ CSCs subpopulations

Based on the above observations, it was suspected that only simultaneously target the two subpopulations to completely eliminate CSCs, so we conducted verification of the antibody-based drug combination therapy. Flow cytometry showed that the anti-ENO1 antibody, MET combined with DDP/CTX significantly reduced the proportion of the ALDH+ and CD44+ subpopulations (Fig. S7). Moreover, the anti-ENO1 antibody, MET combined with DDP/CTX inhibited the self-renewal ability of both CSCs and produced more significant additive inhibitory effects (Fig. 4A). Similarly, remaining living cells, clones, and the proliferation significantly decreased with anti-ENO1 antibody, MET combined with DDP/CTX in two CSCs, indicating that the proliferation potential of CSCs was obviously inhibited (Figs. 4B, 4C and 4D). Besides, the anti-ENO1 antibody, MET combined with DDP/CTX also markedly restrained the migration and invasion abilities of both CSCs (Fig. 4E). Taken together, these data suggested that anti-ENO1 antibody, MET combined with DDP/CTX could simultaneously target ALDH+ and CD44+ subpopulations and display stronger additive inhibition on cell self-renewal, proliferation potential, migration, and invasion abilities, therefore reversing drug resistance induced by CSCs.

Figure 4 Anti-ENO1 antibody, MET combined with DDP/CTX could reverse drug resistance in PAMC82 and A549 CSCs.

(A) The sphere formation assay demonstrated that the self-renewal abilities in MET, anti-ENO1 antibody combined with DDP/CTX-treated PAMC82 and A549 CSCs were both synergistically inhibited. Scale bar, 200 μm. (B, C and D) The colony formation, cell proliferation potential, and CCK-8 assay demonstrated that the cell growth abilities in MET, anti-ENO1 antibody combined with DDP/CTX-treated PAMC82 and A549 CSCs were both inhibited. (E) The cell invasion and migration assay demonstrated that the cell migration and invasion abilities were both significantly inhibited in MET, anti-ENO1 antibody combined with DDP/CTX-treated PAMC82 and A549 CSCs. Scale bar, 200 μm. Data are expressed as mean ± SEM.

Anti-ENO1 antibody, MET combined with DDP/CTX could exert additive inhibition on reversing drug resistance by inhibiting the Wnt/ β-catenin pathway

Given the above results, the internal mechanism of the antibody-based drug combination therapy to reverse drug resistance was further explored. As shown in Fig. 5A, MET alone and MET, anti-ENO1 antibody combined with DDP/CTX could up-regulate the expression of p-AMPK in two CSCs, which was consistent with the results of MET alone or MET combined with DDP/CTX in Fig. 3E, further proving that MET could activate the AMPK pathway to target ALDH+ CSCs subpopulation. Similar results were also observed in Figs. 5A and 3E, that is, the expression of p-AKT decreased with anti-ENO1 antibody alone, and anti-ENO1 antibody, MET combined with DDP/CTX, and anti-ENO1 antibody combined with CTX, further indicating that anti-ENO1 antibody could inhibit the PI3K/AKT pathway. In addition, the increased expression of p-GSK3β was observed with anti-ENO1 antibody, MET combined with DDP/CTX, which further reduced the expression of p-β-catenin and nuclear entry of β-catenin, and then inhibited downstream Cyclin D1 expression (Figs. 5A and 5B). This is consistent with previous studies, that is, in the Wnt/β-catenin pathway, activated GSK3β could phosphorylate β-catenin, causing degradation of β-catenin and inhibiting the accumulation in cells and nuclear, then preventing the transcription of downstream target gene (Cyclin D1), to restrain cell proliferation, migration and other biological effects (Yu et al., 2018). Taken together, these results suggested that the anti-ENO1 antibody, MET combined with DDP/CTX might possess additive inhibition against drug resistance by inhibiting the Wnt/β-catenin pathway.

Figure 5 Anti-ENO1 antibody, MET combined with DDP/CTX could reverse drug resistance and exert additive inhibition by inhibiting the Wnt/β-catenin pathway.

(A) The western blot verified the expression of proteins related to the Wnt/β-catenin pathway in MET, anti-ENO1 antibody combined with DDP/CTX-treated PAMC82 and A549 CSCs. (B) The cell immunofluorescence staining assay demonstrated that the nuclear accumulation of β-catenin was significantly decreased in MET, anti-ENO1 antibody combined with DDP/CTX-treated PAMC82 and A549 CSCs. Scale bar, 30 μm. Data are expressed as mean ± SEM.

Discussion

This study was conducted based on the significance of antibodies as a platform biomaterial against drug resistance caused by CSCs with conventional cancer therapy. To the best of our knowledge, this was the first study that the anti-ENO1 antibody combined with MET could reverse drug resistance with DDP or CTX by targeting both ALDH+ and CD44+ subpopulations by inhibiting the Wnt/β-catenin pathway in GC and NSCLC CSCs (Figs. 6A and 6B).

Figure 6 Anti-ENO1 antibody, MET combined with DDP/CTX could overcome drug resistance caused by CSCs in conventional treatments.

(A) Proposed model for the role of MET, anti-ENO1 antibody combined with DDP/CTX in reversing drug resistance caused by ALDH+ and CD44+ CSCs subpopulations in conventional treatments. (B) Schematic representation of the mechanism by which MET, anti-ENO1 antibody combined with DDP/CTX could reverse drug resistance caused by CSCs in conventional treatments by inhibiting the Wnt/β-catenin pathway.

Increasing studies indicated that following conventional therapy (such as targeted therapy, radio- and chemotherapy) of any tumor, there are residual CSCs (higher invasiveness and chemoresistance) enriched in cancers, which was closely associated with tumor metastasis, recurrence, and drug resistance, and posed inevitable challenges to oncotherapy (Kuşoğlu & Biray Avcı, 2019). The present study found that the proliferation of two cancer cells was significantly inhibited, and the CSCs-like characteristics (such as migration, invasion, proliferation potential, and self-renewal abilities) were not remarkedly restrained with DDP or CTX. In addition, further flow cytometry analysis showed that the proportions of ALDH+ and CD44+ CSCs subpopulations were up-regulated with DDP or CTX. We believed that DDP or CTX alone could cause the enrichment of CSCs subpopulations, then leading to treatment failure or cancer recurrence due to drug resistance, which was similar to previous studies (Phi et al., 2018). Hence, it was reasonably suspected that the therapeutic resistance generated via DDP or CTX could be reversed only by eliminating the CSCs subpopulations.

ENO1 has attracted extensive attention due to its involvement in the regulation of CSCs-like characteristics of cancer cells. Moreover, some studies have found that anti-ENO1 antibody could reduce the proliferation and invasion of cancer cells (Li et al., 2021). This was consistent with our findings that anti-ENO1 antibody could significantly inhibit the proliferation, migration, and invasion of both CSCs. Notably, the proliferation potential and self-renewal ability of CSCs were no statistically significant difference. It was speculated that this might be due to the heterogeneity and plasticity of CSCs as previously reported (Tang, 2012). In other words, epithelial and mesenchymal CSCs subpopulations could dynamically transform their metabolic state to favor glycolysis or oxidative metabolism (Luo et al., 2018; Peiris-Pagès et al., 2016). In parallel, the transformation mechanism is largely responsible for continuous tumor growth, metastasis formation, and recurrence after therapy and is considered a major driver of the development of therapeutic resistance (Shibue & Weinberg, 2017). The proportion of the CD44+ CSCs subpopulation was significantly reduced, while the ALDH+ CSCs subpopulation did not remarkably change with anti-ENO1 antibody in our study, which could further illustrate the above reports and verify the conjecture. Furthermore, it is of great theoretical significance to explore and understand molecular mechanisms. Previous studies indicated that activating the PI3K/AKT pathway might play leading roles in the growth and stemness of EGFR-mediated NSCLC CSCs (Khan et al., 2020). The western blot showed that the expression of p-AKT was down-regulated with the anti-ENO1 antibody. This further indicated that the anti-ENO1 antibody might reduce the CD44+ CSCs subpopulation by inhibiting the PI3K/AKT pathway, which was also demonstrated in previous studies (Wei et al., 2019).

Anti-cancer potential of MET has gained increasing interest due to its inhibition of CSCs (Saini & Yang, 2018), as well as the reports that MET significantly down-regulated the proportion of ALDH+ CSCs subpopulation and enhanced the sensitivity of cancer cells to the DDP in advanced epithelial ovarian cancer (Kim et al., 2019). Similarly, the down-regulation of ALDH+ CSCs subpopulation and inhibition of proliferation potential, proliferation, and self-renewal ability was observed in the present study. In addition, several previous studies have revealed that MET could stimulate AMP-activated protein kinase (AMPK) with higher activities in ALDH+ CSCs subpopulation to possess anti-tumor activity (Guo et al., 2018; Sulaiman et al., 2020). Another study demonstrated that the strong correlation between stemness and drug resistance in GC occurred with the AMPK pathway (Khan et al., 2020). Interestingly, we also found that MET could increase the expression of p-AMPK, which further demonstrated that MET might target the ALDH+ CSCs subpopulation by activating the AMPK pathway. However, MET did not significantly reduce the proportion of CD44+ CSCs subpopulation nor did it inhibit the migration and invasion abilities of CSCs. This may be related to the heterogeneity and plasticity of CSCs similar to the results of anti-ENO1 antibody.

Based on the above studies and observations, it was reasonably assumed that targeting and precisely eradicating all CSCs subpopulations at once could be enough to reverse drug resistance. Thus, the most prominent findings of the present study that the anti-ENO1 antibody, MET combined with DDP/CTX could simultaneously target both CSCs subpopulations and produce stronger additive inhibition to overcome drug resistance, which highlighted the validity of antibody-based drug combination therapy against cancer resistance and the potential as precursor platform of ADC screening. In the present study, anti-ENO1 antibody, MET combined with DDP/CTX not only reduced the proportion of both ALDH+ and CD44+ CSCs subpopulations but also exerted additive inhibition on the proliferation and CSCs-like characteristics (including proliferation potential, self-renewal, migration, and invasion abilities of CSCs). Then, the mechanism validation of these phenotypes attracted further interest.

The western blot showed that the expression of p-AMPK was up-regulated while p-AKT was reduced after the anti-ENO1 antibody, MET combined with DDP/CTX, which further verified that anti-ENO1 antibody and MET could regulate the AMPK and the AKT/PI3K pathway, respectively. Notably, no significant difference in the p-AKT expression of PAMC82 CSCs was found after the anti-ENO1 antibody combined with DDP, while the addition of MET changed the trend. It was speculated that anti-ENO1 antibody might not activate the AMPK enough while MET could strengthen the AMPK pathway so as to regulate the AKT expression. Mounting evidence showed ENO1 could partly inhibit the AMPK pathway in several cancers (Zhan et al., 2017), but no related studies in GC (PAMC82). Moreover, some studies have indicated that activating the AMPK and inhibiting the AKT pathway could enhance the chemical sensitivity to DDP (Gao et al., 2020), and MET could suppress ENO1-AKT1 complex-mediated cell proliferation and EMT signals (Deng et al., 2021). The cases described above were in favor of the surmise.

Strikingly, the increase of p-GSK3β and down-regulation of β-catenin and Cyclin D1 were observed, and cell IF assay showed that β-catenin in the nucleus was decreased. These results indicated that the anti-ENO1 antibody, MET combined with DDP/CTX inactivated phosphorylation of GSK3β, resulting in the decreased accumulation and nuclear entry of β-catenin and downregulation of its target gene Cyclin D1. Accumulating evidence supported that suppressing the Wnt/β-catenin pathway might reverse MDR caused by CSCs and sensitize human cancer to conventional drugs (such as DDP) (Liu et al., 2020; Liu et al., 2019; Yang et al., 2019; Zhu et al., 2020). Besides, anti-ENO1 mAb could attenuate lung cancer cell proliferation, invasion, and metastasis by inhibiting ENO1-mediated GSK3β activation and inactivating the Wnt pathway (Li et al., 2021), and MET might restrain the survival of cancer cells, the stemness of CSCs and EMT by inactivating the Wnt/β-catenin pathway (Conza et al., 2021; Zhang & Wang, 2019). Hence, it was reasonably believed that the inactivation of the Wnt/β-catenin pathway could make it possess stronger additive inhibition and prevent drug resistance related to CSCs. The complex crosstalk and cellular signaling cascades between the PI3K-AKT or AMPK pathway and the Wnt/β-catenin pathway to further promote cancer progression and develop drug resistance have been explored (Khan et al., 2020; Shorning, Dass & Smalley, 2020), but a detailed explanation was missing in this study. Further substantial interest has emerged in the exploration of the interaction between them.

Taken together, the results of the present study have indicated that anti-ENO1 combined with MET could synergistically possess anti-tumor effects. As is well-known, ADC has been considered one of the most effective tools for oncotherapy (Beck et al., 2017). However, identifying cytotoxic molecules suitable as ADC warheads and antibodies reducing attrition rates of drug candidates remains difficult and mandatory (Beck et al., 2017). It seems promising that anti-ENO1 antibody and MET might provide a potential alternative for developing ADC in our study. In addition, taking the above evidence together, anti-ENO1 antibody, MET combined with DDP/CTX might overcome drug resistance caused by the heterogeneity and plasticity of CSCs by inhibiting the Wnt/β-catenin pathway, which suggested that the antibody-based drug combination may be potential and promising for drug resistance. Nevertheless, these findings were drawn from the cell and in vitro model, which is not enough to translate into the complex framework of cancer drug resistance in humans.

Conclusions

The present study revealed that the anti-ENO1 antibody, MET combined with DDP/CTX could effectively possess additive inhibition to reserve drug resistance caused by CSCs enrichment through the Wnt/β-catenin pathway. Notably, anti-ENO1 antibody combined with MET may be a potential precursor platform for ADC screening. In parallel, we propose that antibody-based drug combination therapy (antibody combined with small-molecule drugs) could simultaneously target epithelial and stromal CSCs to effectively reverse the drug resistance in conventional chemotherapy or targeted therapy, which might provide a novel direction of drug repositioning and clinical implications, but efforts are still needed to provide more evidence.

Supplemental Information

Supplemental Information 1 Sphere formation assay demonstrated the self-renewal abilities of PAMC82 cells and A549 cells.

Supplemental Information 2 Western blot for ABCG2 and TOP2A (stemness marker) characterized that PAMC 82 and A549 cells used were CSCs.

Supplemental Information 3 DDP or CTX resulted in the increased expression of ABCG2 (drug efflux transporter) in all two subpopulations of CSCs compared with control group.

Supplemental Information 4 Sphere formation assay demonstrated that the sphere formation ability of PAMC82 CSCs was significantly inhibited by MET at different concentrations, and the self-renewal ability of PAMC82 CSCs was most significantly inhibited by MET at 10 mM.

Supplemental Information 5 Flow cytometry was used to screen antibodies specifically bound to ENO1 on the cell membrane surface and demonstrated that the 18H12 antibody targeting ENO1 could able to recognize ENO1 most accurately.

Supplemental Information 6 Flow cytometry demonstrated that the proportion of the ALDH+ subpopulation was decreased while the proportion of the CD44+ subpopulation was not statistically significant in MET combined with DDP/CTX-treated PAMC82/A549 CSCs; The propo.

Supplemental Information 7 Flow cytometry demonstrated that the proportions of ALDH+ and CD44+ subpopulations were both decreased in MET, anti-ENO1 antibody combined with DDP/CTX-treated PAMC82/A549 CSCS.

Supplemental Information 8 Uncropped Gels/Blots.

Supplemental Information 9 Raw data for cell line experiments.

We appreciate the assistance of Li-Chao Sun in data collection and the assistance of Long Yu in administrative support.

List Of Abbreviations

MET metformin

MDR multiple drug resistance

CSCs cancer stem cells

ADC antibody-drug conjugates

mAb humanized monoclonal antibody

EMT epithelial-to-mesenchymal transition

GC gastric cancer

NSCLC non-small cell lung cancer

DDP cisplatin

CTX cetuximab

EGFR epidermal growth factor receptor

TKIs tyrosine kinase inhibitors

CCK-8 cell counting kit-8

DMEM/F12 Dulbecco’s Modified Eagle Medium/Nutrient Mixture F12

PET polyethylene terephthalate

RIPA radio-immunoprecipitation assay

SDS-PAGE sodium dodecyl sulfate-polyacrylamide gel electrophoresis

IF immunofluorescence

Additional Information and Declarations

Competing Interests

Author Contributions

Data Availability

The authors declare that they have no competing interests.

Xiong Shu conceived and designed the experiments, analyzed the data, prepared figures and/or tables, authored or reviewed drafts of the article, and approved the final draft.

Hui Wen Zhang performed the experiments, analyzed the data, prepared figures and/or tables, authored or reviewed drafts of the article, and approved the final draft.

Shi Ya Liu performed the experiments, prepared figures and/or tables, authored or reviewed drafts of the article, and approved the final draft.

Li Xin Sun performed the experiments, prepared figures and/or tables, authored or reviewed drafts of the article, and approved the final draft.

Tao Zhang performed the experiments, prepared figures and/or tables, authored or reviewed drafts of the article, administrative support, and approved the final draft.

Yu Liang Ran conceived and designed the experiments, prepared figures and/or tables, authored or reviewed drafts of the article, and approved the final draft.

The following information was supplied regarding data availability:

The raw data is available in the Supplemental Files.

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
