# Peer review of "Anti-ENO1 antibody combined with metformin against tumor resistance: a novel antibody-based platform"

_PeerJ, doi:10.7717/peerj.16817_

## Round 0.1 · original submission · Major Revisions

Please address the concerns of all reviewers and revise the manuscript accordingly.

Reviewer 1 ·

Basic reporting

.

Experimental design

.

Validity of the findings

.

Additional comments

Please find the attached file for the review comments.

Annotated reviews are not available for download in order to protect the identity of reviewers who chose to remain anonymous.

Reviewer 2 ·

Basic reporting

In the manuscript, Anti-ENO1 antibody combined with metformin against tumor resistance: a novel antibody-based platform (#83201) the authors have shown that combinatorial treatment of Antibody along-with Metformin could reverse chemoresistance. This study also provides an interesting fact that heterogeneity in Cancer stem cells could be one of the causes of chemoresistance. I recommend the manuscript for publication subject to responses to the following comments.

1) Authors should consider rephrasing the sentence (from line 48 to 51) as it not very clear or try to split the sentence

Experimental design

2) The study would make more sense and would be easy to understand if authors could show the spheroid formation (Spheroidization in Fig.1.B) across days as mentioned in the methods section, as this may represent that spheroid formation was altered initially but after few days it become unchanged due to the self-renewal abilities of cancer stem cells (that would eventually lead to drug resistance).

3) Authors should also consider characterizing the Cancer stem cells by quantifying markers for stemness (like NANOG in normal vs spheroids via qPCR) as well as some chemoresistance genes.

4) In Fig.2 A and B. Authors have shown that DPP and CTX could upregulate ALDH+ and CD44+ CSCs subpopulation. It would be interesting if authors could also look at the expression of drug efflux transporters in these subpopulations of CSCs. Authors should also use a bigger font or try to represent it in a more legible manner.

Validity of the findings

no comment

Additional comments

5) Authors shall also consider putting the name of cell line in Figure.3A (Cell proliferation and IC50 experiment). Please try to split the figure to make it more legible.

Reviewer 3 ·

Basic reporting

The manuscript is well written. no comment.

Experimental design

Material and Methods need to be more detailed. for example, it is not clear how many technical replicates were considered in the cell proliferation, colony formation assay.

Validity of the findings

I doubt the novelty of the study.

Additional comments

Shu et al found that combined treatment of anti-Eno1 and MET could simultaneously target ALDH+ and CD44+ subpopulations and decrease the number of migration, invasion, colon and sphere cells, and proliferation potential, as well as overcome the drug resistance. The results are interesting and significant to the field, although the roles of Eno1 and MET on different cancer types are well-known. The novelty of the study is dramatically decreased by a published paper “ZHANG H, YANG T, YU Z, et al. Anti-ENO1 antibody combined with metformin reverses the resistance of human non-small cell lung cancer A549 cells to cetuximab by targeting cancer stem cells[J]. Chinese Journal of Cancer Biotherapy, 2021: 239-246.”, since they showed similar findings with combined treatment of anti-Eno1 and MET on cancer stem cells. Thus, I would not recommend this paper, unless the authors have further data showing underlying mechanisms by co-treatment of anti-Eno1 and MET or provide vivo data,

---

## Round 0.2 · accepted · Accept

All issues pointed out by the reviewers were adequately addressed and the revised manuscript is acceptable now.

Reviewer 2 ·

Basic reporting

No comments

Experimental design

The statistical analysis in supplementary fig S2, Fig S3 is missing

Validity of the findings

Improved very well

Additional comments

The authors have worked extensively to improve the MS. It can now be published just that authors should do a small correction by including statistical analysis in their new figures in supplementary sections.